# Backprop with Approximate Activations for Memory-efficient Network Training

## Abstract

With innovations in architecture design, deeper and wider neural network models deliver improved performance on a diverse variety of tasks. But the increased memory footprint of these models presents a challenge during training, when all intermediate layer activations need to be stored for back-propagation. Limited GPU memory forces practitioners to make sub-optimal choices: either train inefficiently with smaller batches of examples; or limit the architecture to have lower depth and width, and fewer layers at higher spatial resolutions. This work introduces an approximation strategy that significantly reduces a network's memory footprint during training, but has negligible effect on training performance and computational expense. During the forward pass, we replace activations with lower-precision approximations immediately after they have been used by subsequent layers, thus freeing up memory. The approximate activations are then used during the backward pass. This approach limits the accumulation of errors across the forward and backward pass—because the forward computation across the network still happens at full precision, and the approximation has a limited effect when computing gradients to a layer's input. Experiments, on CIFAR and ImageNet, show that using our approach with 8- and even 4-bit fixed-point approximations of 32-bit floating-point activations has only a minor effect on training and validation performance, while affording significant savings in memory usage.

## 1 Introduction

Deeper neural network models are able to express more complex functions, and recent results have shown that with the use of residual (He et al., 2016a) and skip (Huang et al., 2017) connections to address vanishing gradients, such networks can be trained effectively to leverage this additional capacity. As a result, the use of deeper network architectures has become prevalent, especially for visual inference tasks (He et al., 2016b). The shift to larger architectures has delivered significant improvements in performance, but also increased demand on computational resources. In particular, deeper network architectures require significantly more on-device memory during training—much more so than for inference. This is because training requires retaining the computed activations of all intermediate layers since they are needed to compute gradients during the backward pass.

The increased memory footprint means fewer training samples can fit in memory and be processed as a batch on a single GPU. This is inefficient: smaller batches are not able to saturate all available parallel cores, especially because computation in "deeper" architectures is distributed to be more sequential. Moreover, smaller batches also complicate the use of batch-normalization (Ioffe & Szegedy, 2015), since batch statistics are now computed over fewer samples making training less stable. These considerations often force the choice of architecture to be based not just on optimality for inference, but also practical feasibility for training—for instance, deep residual networks for large images drop resolution early, so that most layers have smaller sized outputs.

While prior work to address this has traded-off memory for computation (Martens & Sutskever, 2012; Chen et al., 2016; Gruslys et al., 2016; Gomez et al., 2017), their focus has been on enabling exact gradient computation. However, since stochastic gradient descent (SGD) inherently works with noisy gradients at each iteration, we propose an algorithm that computes reasonably approximate gradients, while significantly reducing a network's memory footprint and with virtually no additional computational cost. Our work is motivated by distributed training algorithms

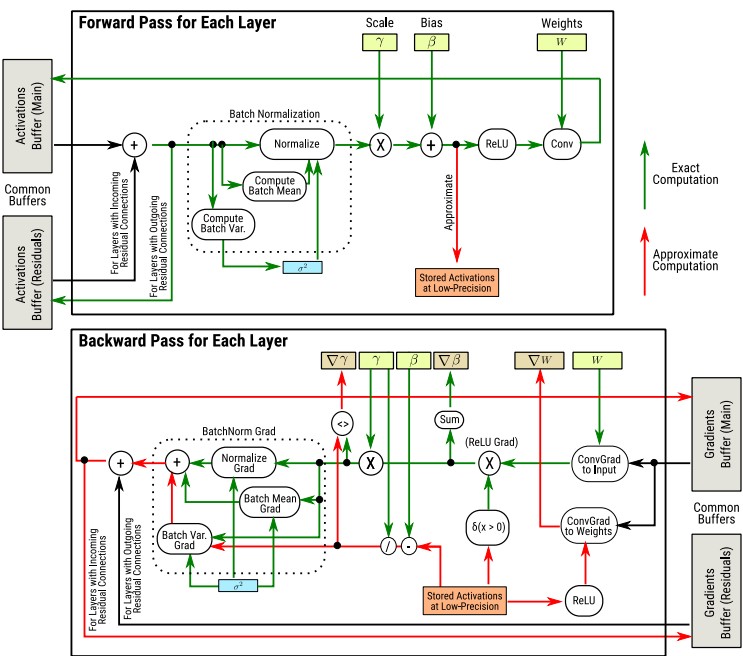

Figure 1: Proposed Approach. We show the computations involved in the forward and backward pass during network training for a single "pre-activation" layer, with possible residual connections. The forward pass is exact, but we discard full-precision activations right after use by subsequent layers (we store these in common global buffers, and overwrite activations once they have been used and no longer needed for forward computation). Instead, we store a low-precision approximation of the activations which occupies less memory, and use these during back-propagation. Our approach limits errors in the gradient flowing back to the input of a layer, and thus accumulation of errors across layers. Since our approximation preserves the signs of activations, most of the computations along the path back to the input are exact—with the only source of error being the use of the approximate activations while back-propagating through the variance-computation in batch-normalization.

that succeed despite working with approximate and noisy gradients aggregated across multiple devices (Recht et al., 2011; Dean et al., 2012; Seide et al., 2014; Wen et al., 2017). We propose using low-precision approximate activations—that require less memory—to compute approximate gradients during back-propagation (backprop) on a single device. Note that training with a lower-precision of 16- instead of 32-bit floating-point representations is not un-common. But this lower precision is used for all computation, and thus allows only for a modest lowering of precision, since the approximation error builds up across the forward and then backward pass through all layers.

In this work, we propose a new backprop implementation that performs the forward pass through the network at full-precision, and incurs limited approximation error during the backward pass. We use the full-precision version of a layer's activations to compute the activations of subsequent layers. However, once these activations have been used in the forward pass, our method discards them and stores a low-precision approximation instead. During the backward pass, gradients are propagated back through all the layers at full precision, but instead of using the original activations, we use their low-precision approximations. As a result, we incur an approximation error at each layer when computing the gradients to the weights from multiplying the incoming gradient with the approximate activations, but ensure the error in gradients going back to the previous layer is minimal.

Our experimental results show that even using only 4-bit fixed-point approximations, for the original 32-bit floating-point activations, causes only minor degradation in training quality. This significantly lowers the memory required for training, which comes essentially for "free"—incurring only the negligible additional computational cost of converting activations to and from low precision representations. Our memory-efficient version of backprop is thus able to use larger batch sizes at each iteration—to fully use available parallelism and compute stable batch statistics—and makes it practical for researchers to explore the use of much larger and deeper architectures than before.

## 2 RELATED WORK

A number of works focus on reducing the memory footprint of a model during inference, e.g., by compression Han et al. (2015) and quantization Hubara et al. (2017), to ensure that it can be deployed on resource-limited mobile devices, while still delivering reasonable accuracy. These methods still require storing full versions of model weights and activations during network training, and assume there is sufficient memory to do so. However, training requires significantly more memory than inference because of the need to store all intermediate activations. And so, memory can be a bottleneck during training, especially with the growing preference for larger and deeper network architectures. A common recourse to this has been to simply use multiple GPUs during training. But, this is inefficient due to the overhead of intra-device communication, and often under-utilizes the available parallelism on each device—computation in deeper architectures is distributed more sequentially and, without sufficient data parallelism, often does not saturate all GPU cores.

A popular strategy to reduce training memory requirements is "checkpointing". Activations for only a subset of layers are stored at a time, and the rest recovered by repeating forward computations (Martens & Sutskever, 2012; Chen et al., 2016; Gruslys et al., 2016). This affords memory savings with the trade-off of additional computational cost—e.g., Chen et al. (2016) propose a strategy that requires memory proportional to the square-root of the number of layers, while requiring up to the computational cost of an additional forward pass. In a similar vein, Gomez et al. (2017) considered network architectures with "reversible" or invertible layers. This allows re-computing intermediate input activations of reversible layers from their outputs during the backward pass.

These methods likely represent the best possible solutions if the goal is restricted to computing exact gradients. But SGD is fundamentally a noisy process, and the exact gradients computed over a batch at each iteration are already an approximation—of gradients of the model over the entire training set (Robbins & Monro, 1985). Researchers have posited that further approximations are possible without degrading training ability, and used this to realize gains in efficiency. For distributed training, asynchronous methods (Recht et al., 2011; Dean et al., 2012) delay synchronizing models across devices to mitigate communication latency. Despite each device now working with stale models, there is no major degradation in training performance. Other methods quantize gradients to two (Seide et al., 2014) or three levels (Wen et al., 2017) so as to reduce communication overhead, and again find that training remains robust to such approximation. Our work also adopts an approximation strategy to gradient computation, but targets the problem of memory usage on a each device. We approximate activations, rather than gradients, with lower-precision representations, and by doing so, we are able to achieve considerable reductions in a model's memory footprint during training. Note that since our method achieves a constant factor saving in memory for back-propagation across any group of layers, it can also be employed within checkpointing to further improve memory cost.

It is worth differentiating our work from those that carry out all training computations at lower-precision Micikevicius et al. (2017); Gupta et al. (2015). This strategy allows for a modest loss in precision: from 32- to 16-bit representations. In contrast, our approach allows for much greater reduction in precision. This is because we carry out the forward pass in full-precision, and approximate activations only after they have been used by subsequent layers. Our strategy limits accumulation of errors across layers, and we are able to replace 32-bit floats with 8- and even 4-bit fixed-point approximations, with little to no effect on training performance. Note that performing all computation at lower-precision also has a computational advantage: due to reduction in-device memory bandwidth usage (transferring data from global device memory to registers) in Micikevicius et al. (2017), and due to the use of specialized hardware in Gupta et al. (2015). While the goal of our work is different, our strategy can be easily combined with these ideas: compressing intermediate activations to a greater degree, while also using 16- instead of 32-bit precision for computation.

## 3 PROPOSED METHOD

### 3.1 BACKGROUND

A neural network is composition of linear and non-linear functions that map the input to the final desired output. These functions are often organized into "layers", where each layer consists of a single linear transformation—typically a convolution or a matrix multiply—and a sequence of non-linearities. We use the "pre-activation" definition of a layer, where we group the linear operation

with the non-linearities that immediately *preceed* it. Consider a typical network whose $l^{th}$ layer applies batch-normalization and ReLU activation to its input $A_{l:i}$ followed by a linear transform:

$$\text{[Batch Norm.]} \quad A_{l:1} = (\sigma_l^2 + \epsilon)^{-1/2} \circ (A_{l:i} - \mu_l), \mu_l = \text{Mean}(A_{l:i}), \sigma^2 = \text{Var}(A_{l:i}); \quad (1)$$

$$\text{[Scale \& Bias]} \quad A_{l:2} = \gamma_l \circ A_{l:1} + \beta_l; \quad (2)$$

$$\text{[ReLU]} \quad A_{l:3} = \max(0, A_{l:2}); \quad (3)$$

$$\text{[Linear]} \quad A_{l:o} = A_{l:3} \times W_l; \quad (4)$$

to yield the output activations $A_{l:o}$ that are fed into subsequent layers. Here, each activation is a tensor with two or four dimensions: the first indexing different training examples, the last corresponding to "channels", and others to spatial location. $\text{Mean}(\cdot)$ and $\text{Var}(\cdot)$ aggregate statistics over batch and spatial dimensions, to yield vectors $\mu_l$ and $\sigma_l^2$ with *per-channel* means and variances. Element-wise addition and multiplication (denoted by $\circ$) are carried out by "broadcasting" when the tensors are not of the same size. The final operation represents the linear transformation, with $\times$ denoting matrix multiplication. This linear transform can also correspond to a convolution.

Note that (1)-(4) are defined with respect to learnable parameters $\gamma_l, \beta_l$, and $W_l$, where $\gamma_l, \beta_l$ are both vectors of the same length as the number of channels in $A_l$, and $W_l$ denotes a matrix (for fully-connected layers) or elements of a convolution kernel. These parameters are learned iteratively using SGD, where at each iteration, they are updated based on gradients—$\nabla\gamma_l, \nabla\beta_l$, and $\nabla W_l$—of some loss function with respect to these parameters, computed on a batch of training samples.

To compute gradients with respect to all parameters for all layers in the network, the training algorithm first computes activations for all layers in sequence, ordered such that each layer in the sequence takes as input the output from a previous layer. The loss is computed with respect to activations of the final layer, and then the training algorithm goes through all layers again in reverse sequence, using the chain rule to back-propagate gradients of this loss. For the $l^{th}$ layer, given the gradients $\nabla A_{l:o}$ of the loss with respect to the output, this involves computing gradients $\nabla\gamma_l, \nabla\beta_l$, and $\nabla W_l$ with respect to the layer's learnable parameters, as well as gradients $\nabla A_{l:i}$ with respect to its input for further propagation. These gradients are given by:

$$\text{[Linear]} \quad \nabla W = A_{l:3}^T \times (\nabla A_{l:o}), \quad \nabla A_{l:3} = (\nabla A_{l:o}) \times W_l^T; \quad (5)$$

$$\text{[ReLU]} \quad \nabla A_{l:2} = \delta(A_{l:2} > 0) \circ (\nabla A_{l:3}); \quad (6)$$

$$\text{[Scale \& Bias]} \quad \nabla\beta_l = \text{Sum}(\nabla A_{l:2}), \quad \nabla\gamma_l = \text{Sum}\left(A_{l:1} \circ (\nabla A_{l:2})\right), \quad \nabla A_{l:1} = \gamma_l \circ \nabla A_{l:2}; \quad (7)$$

$$\text{[Batch Norm.]} \quad \nabla A_{l:i} = (\sigma_l^2 + \epsilon)^{-1/2} \circ \left[\nabla A_{l:1} - \text{Mean}(\nabla A_{l:1}) - A_{l:1} \circ \text{Mean}(A_{l:1} \circ \nabla A_{l:1})\right]; \quad (8)$$

where $\text{Sum}(\cdot)$ and $\text{Mean}(\cdot)$ involve aggregation over all but the last dimension, and $\delta(A > 0)$ a tensor the same size as $A$ that is one where the values in $A$ are positive, and zero otherwise.

When the goal is to just compute the final output of the network, the activations of an intermediate layer can be discarded during the forward pass as soon as we finish processing the subsequent layer or layers that use it as input. However, we need to store all these intermediate activations during training because they are needed to compute gradients during back-propagation: (5)-(8) involve not just the values of the incoming gradient, but also the values of the activations themselves. Thus, training requires enough available memory to hold the activations of all layers in the network.

## 3.2 BACK-PROPAGATION WITH APPROXIMATE ACTIVATIONS

We begin by observing we do not necessarily need to store all intermediate activations $A_{l:1}, A_{l:2}$, and $A_{l:3}$ *within* a layer. For example, it is sufficient to store the activation values $A_{l:2}$ right before the ReLU, along with the variance vector $\sigma_l^2$ (which is typically much smaller than the activations themselves). Given $A_{l:2}$, we can reconstruct the other activations $A_{l:3}$ and $A_{l:3}$ needed in (5)-(8) using element-wise operations, which typically have negligible computational cost compared to the linear transform itself. Some deep learning frameworks already use such "fused" layers to conserve memory, and we consider this to be our "baseline" for memory usage.

However, storing one activation tensor at full-precision for every layer still requires a considerable amount of memory. We therefore propose retaining an approximate low-precision version $\tilde{A}_{l:2}$ of $A_{l:2}$, that requires much less memory for storage, for use in (5)-(8) during back-propagation.

As shown in Fig. 1, we use full-precision versions of all activations during the forward pass to compute $A_{l:o}$ from $A_{l:i}$ as per (1)-(4), and use $A_{l:2}$ to compute its approximation $\tilde{A}_{l:2}$. The full precision approximations are discarded as soon they have been used—the intermediate activations $A_{l:1}, A_{l:2}, A_{l:3}$ are discarded as soon as the approximation $\tilde{A}_{l:2}$ and output $A_{l:o}$ have been computed, and $A_{l:o}$ is discarded after it has been used by a subsequent layer. Thus, only the approximate activations $\tilde{A}_{l:2}$ and (full-precision) variance vector $\sigma_l^2$ are retained in memory for back-propagation.

We use a simple, computationally inexpensive approach to approximate $A_{l:2}$ via a $K$-bit fixed-point representation for some desired value of $K$. Since $A_{l:1}$ is normalized to be zero-mean and unit-variance, $A_{l:2}$ has mean $\beta_l$ and variance $\gamma_l^2$. We compute an integer tensor $\tilde{A}_{l:2}^*$ from $A_{l:2}$ as:

$$\tilde{A}_{l:2}^* = \text{Clip}_K( \ \lfloor A_{l:2} \circ 2^K (6 * \gamma_l)^{-1} \rfloor + 2^{K-1} - \lfloor \beta_l \circ 2^K (6 * \gamma_l)^{-1} \rfloor), \tag{9}$$

where $\lfloor \cdot \rfloor$ indicates the "floor" operator, and $\text{Clip}_K(x) = \max(0, \min(2^K - 1, x))$. The resulting integers (between 0 and $2^K - 1$) can be directly stored with $K$-bits. When needed during back-propagation, we recover a floating-point tensor holding the approximate activations $\tilde{A}_{l:2}$ as:

$$\tilde{A}_{l:2} = 2^{-K}(6 * \gamma_l) \circ (\tilde{A}_{l:2}^* + 0.5 - 2^{K-1} + \lfloor \beta_l \circ 2^K (6 * \gamma_l)^{-1} \rfloor). \tag{10}$$

This simply has the effect of clipping $A_{l:2}$ to the range $\beta_l \pm 3\gamma_l$ (approximately, the range may be slightly asymmetric around $\beta_l$ because of rounding), and quantizing values in $2^K$ fixed-size intervals (to the median of each interval) within that range. However, crucially, this approximation ensures that the sign of each value is preserved, i.e., $\delta(A_{l:2} > 0) = \delta(\tilde{A}_{l:2} > 0)$.

### 3.3 Approximation Error in Training

Since the forward computations happen in full-precision, there is no error introduced in any of the activations $A_l$ prior to approximation. To analyze the error introduced by our approach, we then consider the effect of using $\tilde{A}_{l:2}$ instead of $A_{l:2}$ (and equivalently, $\tilde{A}_{l:1}$ and $\tilde{A}_{l:3}$ derived from $\tilde{A}_{l:2}$) to compute gradients in (5)-(8). We begin by noting that for all values of $A_{l:2}$ that fall within the range $\beta_l \pm 3\gamma_l$ (and are therefore not clipped), the worst-case approximation error in the activations themselves is bounded by half the width of the quantization intervals:

$$|A_{l:2} - \tilde{A}_{l:2}| \le {}^3/2^K \gamma = {}^3/2^K \sqrt{\text{Var}(A_{l:2})}, \tag{11}$$

where $\text{Var}(\cdot)$ denotes per-channel variance (and the RHS is interpreted as applying to all channels). Hence, the approximation error is a fraction of the variance in the activations themselves, and is lower for higher values of $K$. It is easy to see that $|A_{l:3} - \tilde{A}_{l:2}| \le |A_{l:2} - \tilde{A}_{l:3}|$ since $A_{l:3}$ and $\tilde{A}_{l:3}$ are derived from $A_{l:2}$ and $\tilde{A}_{l:3}$ by clipping negative values of both to 0, which only decreases the error. Further, since $A_{l:2}$ is related to $A_{l:1}$ by simply scaling, the error in $\tilde{A}_{l:1}$ is also bounded as a fraction of *its* variance, which is one, i.e: $|A_{l:1} - \tilde{A}_{l:1}| \le {}^3/2^K$.

We next examine how these errors in the activations affect the accuracy of gradient computations in (5)-(8). During the first back-propagation step in (5) through the linear transform, the gradient $\nabla W$ to the learnable transform weights will be affected by the approximation error in $\tilde{A}_{l:3}$. However, the gradient $\nabla A_{l:2}$ can be computed exactly (as a function of the incoming gradient to the layer $\nabla A_{l:o}$), because it does not depend on the activations. Back-propagation through the ReLU in (7) is also not affected, because it depends only on the sign of the activations, which is preserved by our approximation. When back-propagating through the scale and bias in (6), only the gradient $\nabla \gamma$ to the scale depends on the activations, but gradients to the bias $\beta_l$ and to $A_{l:1}$ can be computed exactly.

And so, although our approximation introduces some error in the computations of $\nabla W$ and $\nabla \gamma$, there is no error introduced in the gradient flowing towards the input of the layer, until it reaches the batch-normalization operation in (8). Here, we do incur an error, but note that this is only in one of the three terms of the expression for $\nabla A_{l:i}$—which accounts for back-propagating through the variance computation, and is the only term that depends on the activations. Hence, while our activation approximation does introduce some errors in the gradients for the learnable weights, we limit the accumulation of these errors across layers because a majority of the computations for back-propagation to the input of each layer are exact. This is illustrated in Fig. 1, with the use of green arrows to show computations that are exact, and red arrows for those affected by the approximation.

### 3.4 Network Architectures and Memory Usage

Our full training algorithm applies our approximation strategy to every layer (defined by grouping linear transforms with preceding non-linear activations) during the forward and backward pass. Skip and residual connections are handled easily, since back-propagation through these connections involves simply copying to and adding gradients from both paths, and doesn't involve the activations themselves. (Although we do not consider this in our implementation, older residual connections that are added after batch-normalization but before the ReLU can also be handled, but would require saving activations both before and after addition—in the traditional case, well as our approach).

Our method is predicated on the use of ReLU activations since its gradient depends only on the sign of the activations, and can be used for other such non-linearities such as "leaky"-ReLUs. Other activations (like sigmoid) may incur additional errors—in particular, we do not approximate the activations of the final output layer in classifier networks that go through a Soft-Max. However, since this is typically at the final layer, and computing these activations is immediately followed by back-propagating through that layer, approximating these activations offers no savings in memory. Our approach also handles average pooling by simply folding it in with the linear transform. For max-pooling, exact back-propagation through the pooling operation would require storing the arg-max indices (the number of bits required to store these would depend on the max-pool receptive field size). However, since max-pool layers are used less often in recent architectures in favor of learned downsampling (ResNet architectures for image classification use max-pooling only in one layer), we instead choose not to approximate layers with max-pooling for simplicity.

Given a network with $L$ layers, our memory usage depends on connectivity for these layers. Our approach requires storing the approximate activations for each layer, each occupying reduced memory rate at a fractional rate of $\alpha < 1$. During the forward pass, we also need to store, at full-precision, those activations that are yet to be used by subsequent layers. This is one layer's activations for feed-forward networks, and two layers' for standard residual architectures. More generally, we will need to store activations for upto $W$ layers, where $W$ is the "width" of the architecture—which we define as the maximum number of outstanding layer activations that remain to be used as process layers in sequence. During back-propagation, the same amount of space is required for storing gradients till they are used by previous layers. We also need space to re-create a layer's approximate activations as full-precision tensors from the low-bit stored representation, for use in computation.

Thus, assuming that all activations of layers are the same size, our algorithm requires $\mathcal{O}(W+1+\alpha L)$ memory, compared to the standard requirement of $\mathcal{O}(L)$. This leads to substantial savings for deep networks with large $L$ (note $\alpha = \frac{1}{4}, \frac{1}{8}$ when approximating 32-bit floats with $K = 8, 4$ bits).

## 4 Experiments

We developed a library that implements the proposed method for approximate memory-efficient training, given a network architecture specification which can include residual layers (i.e., $W = 2$). As illustrated in Fig. 1, the method allocates a pair of global buffers for the direct and residual paths that is common to all layers. At any point during the forward pass, these buffers hold the full-precision activations that are needed for computation of subsequent layers. The same buffers are used to store gradients during the back-ward pass. Beyond these common buffers, the library only stores the low-precision approximate activations for each layer for use during the backward-pass. Further details of our implementation are provided in the appendix. We compare our approximate training approach, with 8- and 4-bit activations, to exact training with full-precision activations as a baseline. For a fair comparison, we again only store one set of activations like our method for a group of batch-normalization, ReLU, and linear (convolution) operations. This is achieved with our library by storing $A_{l:2}$ without approximation ($\tilde{A}_{l:2} = A_{l:2}$).

**CIFAR-10 and CIFAR-100.** We begin with comparisons on 164-layer pre-activation residual networks (He et al., 2016b) on CIFAR-10 and CIFAR-100 (Krizhevsky & Hinton, 2009), using three-layer "bottlneck" residual units and parameter-free shortcuts for all residual connections. We train the network for 64k iterations with a batch size of 128, momentum of 0.9, and weight decay of $2 \times 10^{-4}$. Following He et al. (2016b), the learning rate is set to $10^{-2}$ for the first 400 iterations, then increased to $10^{-1}$, and dropped by a factor of 10 at 32k and 48k iterations. We use standard data-augmentation with random translation and horizontal flips. We train these networks with our

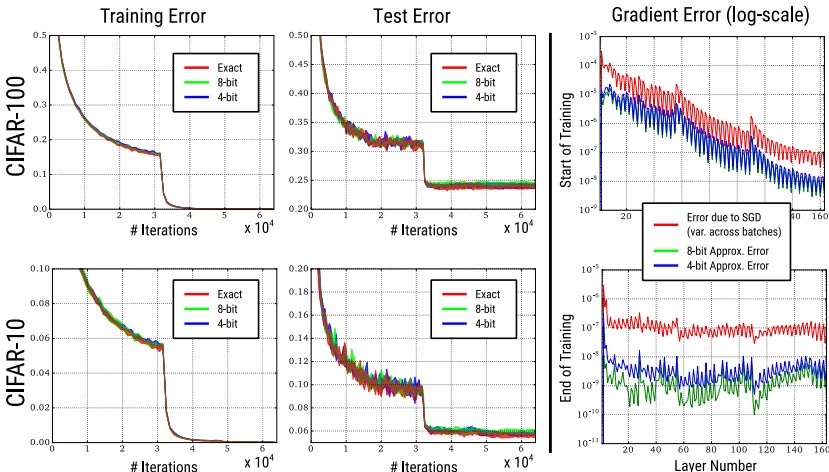

Figure 2: Approximate Training on CIFAR. **(Left)** We show the evolution of training and test error for ResNet-164 models trained on CIFAR-10 and CIFAR-100 (with four different random seeds for each case) and find performance of approximate training closely follows that of the exact baseline. **(Right)** We visualize errors in the computed gradients of learnable parameters (convolution kernels) for different layers for two snapshots of a CIFAR-100 model at the start and end of training. We plot errors between the true gradients and those computed by our approximation, averaged over a 100 batches. We compare to the errors from SGD itself: the variance between the (exact) gradients computed from different batches, and find this to be 1-2 orders of magnitude higher.

Table 1: Accuracy Comparisons on CIFAR and ImageNet. On CIFAR-10 and CIFAR-100, we report test set error statistics with ResNet-164 over four models trained with different random seeds. For ImageNet, we report 10-crop Top-5 error on the validation set with ResNet-34 and ResNet-152.

|  | **CIFAR-10** ($L = 164$) | | **CIFAR-100** ($L = 164$) | | **ImageNet** | |
| --- | --- | --- | --- | --- | --- | --- |
|  | Median | Mean $\pm$ STD | Median | Mean $\pm$ STD | ($L = 34$) | ($L = 152$) |
| Exact ($\alpha = 1$) | 5.56% | 5.54% $\pm$ 0.14 | 23.59% | 23.58% $\pm$ 0.35 | 10.06% | 7.20% |
| 8-bit ($\alpha = 1/4$) | 5.61% | 5.63% $\pm$ 0.14 | 23.63% | 23.75% $\pm$ 0.39 | 10.60% | 7.70% |
| 4-bit ($\alpha = 1/8$) | 5.63% | 5.62% $\pm$ 0.07 | 23.66% | 23.71% $\pm$ 0.29 | 10.74% | 7.72% |

approach using $K = 8$ and $K = 4$ bit approximations, and measure degradation in accuracy with respect to the baseline—repeating training for all cases with four random seeds. We visualize the evolution of training and test set error in Fig. 2, and report statistics of the final test error in Table 1.

We find that both training and test errors when using our low-memory approximation strategy closely follow those of exact back-propagation, throughout the training process. Moreover, the final median test errors of models trained with even 4-bit approximations (i.e., corresponding to $\alpha = 1/8$) are higher only by $0.07\%$ compared to those trained with exact computations. To examine the reason behind this robustness, Fig. 2 also visualizes the error in the final *parameter* gradients used to update the model. Specifically, we take two models for CIFAR-100—at the start and end of training—and then compute gradients for a 100 batches with respect to the convolution kernels of all layers exactly, and using our approximate strategy. We plot the average squared error between these gradients. We compare this approximation error to the "noise" inherent in SGD, due to the fact that each iteration considers a random batch of training examples. This is measured by average variance between the (exact) gradients computed in the different batches. We see that our approximation error is between one and two orders of magnitude below the SGD noise for all layers, both at the start and end of training. So while we do incur an error due to approximation, this is added to the much higher error that already exists due to SGD even in exact training, and hence further degradation is limited.

**ImageNet.** We also report results on training models for ImageNet (Russakovsky et al., 2015). Here, we consider two residual architectures: 34-layer (with two-layer units without bottlenecks) and 152-layer (with three-layer bottleneck units)—again using pre-activation parameter-free shortcuts. We

Table 2: Comparison of maximum batch-size and wall-clock time per training example (i.e., training time per-iteration divided by batch size) for different ResNet architectures on CIFAR-10.

|  |  | ResNet-164 | ResNet-254 | ResNet-488 | ResNet-1001 | ResNet-1001-4x |
|---|---|---|---|---|---|---|
| Maximum | Exact | 688 | 474 | 264 | 134 | 26 |
| Batch-size | 4-bit | 2522 | 2154 | 1468 | 876 | 182 |
| Run-time | Exact | 4.1 ms | 6.5 ms | 13.3 ms | 31.3 ms | 130.8 ms |
| per Sample | 4-bit | 4.3 ms | 6.7 ms | 12.7 ms | 26.0 ms | 101.6 ms |

train with a batch size of 256 for a total of 640k iterations with a momentum of 0.9, weight decay of $10^{-4}$, and standard scale, color, flip, and translation augmentation. The initial learning rate is set to $10^{-1}$ with drops by factor of 10 every 160k iterations. Table. 1 reports top-5 validation accuracy (using 10 crops at a scale of 256) for models trained using exact computation, and our approach with $K = 8$ and $K = 4$ bit approximations. Again, the drop in accuracy is relatively small: at $0.7\%$ and $0.5\%$ for the 34- and 152-layer models respectively, for a memory savings factor of $\alpha = 1/8$.

**Memory and Computational Efficiency.** For the CIFAR experiments, the full 128-size batch fit on a single 1080Ti GPU for both the baseline and our method. For ImageNet with ResNet-34, our method could fit the 256-sized batch, but not the baseline—for which we used two passes with 128-size batches and averaged the gradients for each iteration. For ResNet-152, we parallelized computation across two GPUs, and again, our method could fit half a batch (size 128) on each GPU, the baseline required two passes with $64-$sized batches per-GPU per-pass. In the CIFAR experiments and ImageNet with ResNet-34, the batches were large enough to saturate all GPU cores for both our method and the baseline. In this case, the running times per iteration were almost identical—with a very slight increase in our case due to the cost computing approximations: exact vs approximate (4-bit) training took 0.66 seconds vs 0.72 seconds for CIFAR-100, and 1.68 seconds vs 1.71 seconds for ImageNet ResNet-34. But for ResNet-152 on ImageNet, the 64-sized batch for exact training underutilized the available parallelism, and the time per-iteration (across two GPUs) was 2s vs 1.7s for exact vs approximate (4-bit) training.

However, these represent comparisons restricted to have the same total batch size (needed to evaluate relative accuracy). For a more precise evaluation of memory usage, and the resulting computational efficiency from parallelism, we considered residual networks for CIFAR-10 of various depths up to 1001 layers—and additionally for the deepest network, a version with four times as many feature channels in each layer. For each network, we measured the largest batch size that could be fit in memory with our method (with $K = 4$) vs the baseline, i.e., $b$ such that a batch of $b + 1$ caused an out-of-memory error on a 1080Ti GPU. We also measured the corresponding wall-clock training time per sample, computed as the training time per-iteration divided by this batch size. These results are summarized in Table 2. We find that in all cases, our method allows significantly larger batches to be fit in memory. Moreover, for larger networks, our method also yields a notable computational advantage since the larger batches permit full exploitation of available cores on the GPU.

## 5 CONCLUSION

We introduced a new algorithm for approximate gradient computation in neural network training, that significantly reduces the amount of required on-device memory. Our experiments show that this comes at a minimal cost in terms of both quality of the learned models, and computational expense. With a lower memory footprint, our method allows training with larger batches in each iteration—improving efficiency and stability—and exploration of deeper architectures that were previously impractical to train. We will release our reference implementation on publication.

Our method shows that SGD is reasonably robust to working with approximate activations. While we used an extremely simple approximation strategy—uniform quantization—in this work, we are interested in exploring whether more sophisticated techniques—e.g., based on random projections or vector quantization—can provide better trade-offs, especially if informed by statistics of gradients and errors from prior iterations. We are also interested in investigating whether our approach to partial approximation can be utilized in other settings, especially to reduce inter-device communication for distributed training with data or model parallelism.

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

## APPENDIX: IMPLEMENTATION DETAILS

We implemented our approximate training algorithm using the TensorFlow library (Abadi et al., 2016). However, we only used TensorFlow's functions for individual forward and gradient computations, but not on its automatic differentiation functionality. Instead, our library allows specifying general residual network architectures, and based on this specification, creates a set of TensorFlow ops for doing forward and backward passes through each layer. We also used custom ops implemented in CUDA to handle the conversion to and from low-precision representations. Each layer's forward and backward ops are called in separate `sess.run` calls, and all data that needs to persist between calls—including still to be used full precision activations and gradients in the forward and backward pass, and approximate intermediate activations—are stored explicitly as Tensorflow variables.

For the forward and backward passes through the network, we call these ops in sequence followed by ops to update the model parameters based on the computed gradients. We chose not to allocate and then free variables for the full-precision layer activations and gradients, since this caused memory fragmentation with Tensorflow's memory management routines. Instead, as described in Sec. 4, we used two common variables as buffers for all layers to hold activations (in the forward pass) and gradients (in the backward pass) for the direct and residual paths in the network respectively. We reuse these buffers by overwriting old activations and gradients with new ones. The size of these buffers is set based on the largest layer, and we used slices of these buffers for smaller layers.

