# OpenReview forum: "Backprop with Approximate Activations for Memory-efficient Network Training"
_ICLR.cc/2019/Conference_

### Official Review · AnonReviewer2 · 2018-10-31
**Borderline paper**

**Rating:** 5
**Confidence:** 5

**Review:**

This paper proposes to use 8/4-bit approximation of activations to save the memory cost during gradient computation.  The proposed technique is simple and straightforward. On the other hand, the proposed method only saves up to a constant cost of the storage. With the constant factor (4x, 8x) depending on whether fp16 or fp32 is used during computation. Notably, there is a small but noticeable accuracy drop in the final trained model using this mechanism.

The alternative method, gradient checkpointing, can bring sublinear memory improvement, with at most 25%  compute overhead, with no loss of accuracy drop.

As a result, the proposed method has a limited use case. The author did mention, during the response that the method could be combined further with the sublinear checkpointing. However, since sublinear checkpointing already brings in significant savings, it is unclear whether low bit compression is necessary.

Given the limited technical novelty(can be described as oneliner "store forward pass in 4/8 bit fixed point"),  limited applicable scenarios, and limited improvement it can buy(4x memory saving with accuracy drop), I think this is a boarder-line paper

On the positive side, the empirical result could still be interesting to some readers in the ICLR community, the paper could be further improved by comparing more numerical representations, such as fp16 and other floating point formats such as unum.

---

> ### Author Response · Authors · 2018-11-08
> **Clarifications**
>
> - There has been a misunderstanding. We would like to clarify that the baseline (without approximation) that we compare to does in fact include the "cheap" form of checkpointing that the reviewer suggests (we agree it is the right baseline to compare to). We indicate this in the end of the first para of Sec 3.2 and in the third para of Sec 4---but we realize that it could have been stated more clearly (i.e., without notation), and we shall do so in the revised version.
>
> Thus, our baseline in fact does only store one set of activations for the set of batchnorm, relu, and conv (we count this as one layer in our definition of L). And our approximation strategy provides us a saving (of ~ 4x to 8x) over and above those of this basic form of checkpointing. Instead of storing that one set of activations in full floating point precision, we approximate it (after using the full precision version for the forward pass).
>
> - Also we want to clarify that compared to the more expensive forms of checkpointing---which permit sub-linear memory usage but require expensive recomputation of groups of conv layers--our method is nearly free in terms of computation cost---the only additional cost is elementwise rounding of activations, which is relatively negligible in a typical network as noted in the experiments. And even though our savings are linear, we believe a factor of 4x or 8x savings with nearly identical computational cost can be extremely useful in many settings.
>
> Moreover, in cases where memory is especially at a premium, our approximation-based method can be _combined_ with checkpointing. When breaking up the network into groups of layers, our method can be used to reduce the memory footprint even further for back-propagating within each group, thus allowing larger groups, fewer checkpoints, and hence less computational cost for the same memory budget. Essentially, our strategy is orthogonal (and therefore, potentially complementary) to checkpointing.

---

> > ### Comment · AnonReviewer2 · 2018-11-19
> > **RE: Clarifications**
> >
> > Thanks for the clarification, I have modified my reviews accordingly.

---

> > > ### Author Response · Authors · 2018-11-19
> > > **RE: Clarifications**
> > >
> > > Thanks for the update! But, we would like to emphasize that our method is not an alternative, but **complementary**, to gradient checkpointing. One doesn’t preclude the use of the other.
> > >
> > > Checkpointing saves memory by dropping some activations and recomputing them during the backward pass. We reduce the amount of memory needed to save each set of activations. And so, within checkpointing, our method will require fewer activations to be dropped and recomputed, in turn improving the computational overhead of checkpointing.
> > >
> > > Thus, checkpointing and our method are both ways of reducing the memory footprint of training,  each with their own trade-off. More importantly, they can be used by themselves or together. And since state-of-the-art networks for most tasks are only getting deeper, researchers and practitioners will be interested in exploiting all possible avenues for saving memory. This is why we are certain our method will be practically useful in many cases.
> > >
> > > Gradient checkpointing is an elegant solution, and likely the best possible one for exact computation---as we already say in our paper (third para, Sec 2). In the revised version, we will clarify that our method is complementary to it.

---

> > > > ### Author Response · Authors · 2018-11-19
> > > > **RE: Clarifications**
> > > >
> > > > To further clarify, let’s consider Chen et al. 2016’s checkpointing approach.
> > > >
> > > > Following their derivation (based on a feed-forward network), they divide n layers into k segments of n/k layers each. Their memory cost (eq 1 in their paper)  is O(n/k)+O(k), where the first term refers to the memory needed to do forward-backward on each segment (one at a time). The optimal k that minimizes this is sqrt(n), and so memory cost is O(sqrt(n)).
> > > >
> > > > In our case, our method is applied for back-propagating within each segment to reduce per-segment memory cost, we would have O(An/k)+O(k) (where A is the \alpha in our paper = ¼ or ⅛). Now the optimal k = sqrt(An), and so memory cost is only O(sqrt(An)). Thus our factor improvement in memory cost carries over (within the sqrt).
> > > >
> > > > The additional computation cost corresponds to the repeated forward computation for all but the last segment and checkpointed layers. So, O(n-n/k-k). In regular checkpointing, this is O(n-2*sqrt(n)). In our case, it will be O(n-(1/sqrt(A)+sqrt(A))*sqrt(n)), again an improvement.
> > > >
> > > > Thus incorporating our method provides further benefits over and above checkpointing---in both memory and computation.

---

> > > > > ### Comment · AnonReviewer2 · 2018-11-19
> > > > > **RE: Clarifications**
> > > > >
> > > > > Thanks for the clarification, I have updated my reviews. I think the empirical results have a certain value, I would certainly like the result if it as an ICLR workshop paper or an arxiv.
> > > > >
> > > > > But it is still a boarder-line paper due to the limited novelty in the techniques being proposed and limited improvements it can buy.
> > > > >
> > > > > One way to improve the paper is to provide a more extensive study of numeric format(e.g. fp16, unums).

---

> > > > > > ### Author Response · Authors · 2018-11-19
> > > > > > **RE: Clarifications**
> > > > > >
> > > > > > Thanks! One quick comment---while we recognize that technical novelty is a subjective evaluation, we'd like to point out that there is non-trivial aspect to our approach that is new and goes beyond simple quantization.
> > > > > >
> > > > > > We don't quantize activations right away, but only **after** they have been used by subsequent layers in the forward pass. This is key in ensuring the forward pass is computed at full precision, and that the errors in the backward pass and to weight gradients are limited and do not accumulate.

---

### Official Review · AnonReviewer1 · 2018-11-02
**Clear explanation and execution of good idea**

**Rating:** 7
**Confidence:** 4

**Review:**

The authors detail a procedure to reduce the memory footprint of deep networks by quantization of the activations only on back propagation. While this scheme does not benefit from computational speedups of activation quantization on both passes (and indeed has a slight computational overhead), the authors demonstrate that for common convolutional architectures it nicely preserves the accuracy of computation by computing the forward pass at full accuracy and limiting propagation of errors in the backward pass. This is possible because the majority of errors are introduced in gradient calculation of the weights and not the inputs each layer. The authors also wisely perform quantization after batch normalization and use the known mean and variance of the activations to scale the quantization and reduce errors. They demonstrate very slight drops in performance accuracy for ResNets on Cifar10, Cifar100, and ImageNet with memory compression factors up to 8. They also point to natural future directions such as using vector quantization to better leverage the activation statistics. The paper is also very clearly written with appropriate references to the relevant literature.

An area of improvement I could see for the paper would be to demonstrate the utility of the reduced memory footprint. Their motivation clearly outlines that reducing memory can allow for larger batch sizes and larger networks that can improve the performance of training, but the authors do not demonstrate an example of this principle. They do mention that they are able to train with a larger batch size on ImageNet without combining batches, but more quantitative evidence of improvements in wall clock time (for different batch sizes) or improvement in performance (for larger networks) would help support the arguments of the paper. Given that the authors are focusing on single device training, they don't have to necessarily improve the state of the art, but a relative comparison would be illustrative. Also, specific measurements of the change in memory footprint for real networks would be helpful.

---

> ### Author Response · Authors · 2018-11-11
> **Response**
>
> Thank you for your encouraging comments and suggestions.
>
> - Based on the reviewer's suggestion, we ran an experiment to obtain an (indirect) measurement of real memory usage of our method.  This was done by searching for the maximum batch size that could be fit in memory on a single GPU (i.e., b such that b+1 causes an out of memory error). We also measured the training times per sample---by measuring the wall clock training times per iteration and dividing it by the respective batch sizes.
>
> We did this for both the baseline (i.e., no approximation) and our approach with 4-bit quantization with Resnets on CIFAR-10 with increasing number of layers and, for the deepest network, a version with 4x feature channels for all intermediate layers. The results are as follows:
>
> All numbers are Baseline vs 4-bit Approximation, in that order.
>
> Resnet-1001-4x: [Max Batch Size: 26  vs 182   ]  [Time per sample: 130.8 ms vs 101.6 ms]
> Resnet-1001:      [Max Batch Size: 134 vs 876  ]  [Time per sample:  31.3 ms vs  26.0 ms]
> Resnet-488:        [Max Batch Size: 264 vs 1468]  [Time per sample:  13.3 ms vs  12.7 ms]
> Resnet-254:        [Max Batch Size: 474 vs 2154]  [Time per sample:   6.5 ms vs   6.7 ms]
> Resnet-164:        [Max Batch Size: 688 vs 2582]  [Time per sample:   4.1 ms vs   4.3 ms]
>
> Thus our method allows significantly larger batches to be fit in memory. These are actual gains from our implementation, which will be released publicly with the paper. Moreover, for larger networks, our method provides us an advantage in wall-clock time. This is because the computation becomes memory bound when using lower batch sizes with regular training, and not all GPU cores are saturated. For smaller networks where the baseline is able to fit in a large enough batch to saturate the GPU, we have a small increase in the time.  This increase corresponds to time for computing the approximation.
>
> We sincerely thank the reviewer for this suggestion. We believe these experimental results give readers tangible numbers that illustrate the benefits of using our approach in practice. We will add them to the paper.
>
>
> -Multi-GPU Training: Our implementation also supports multi-GPU training with data parallelism (i.e., splitting batches across GPUs). Here, our approximation allows for lower memory and therefore larger batches on each GPU. Note that the time per sample metric also applies to multi-GPU training, where it corresponds to time per sample per GPU. Thus, for a fixed number of GPUs, the wall-clock time advantage of our method for larger networks carries over.
>
> Since the original submission, we have run an experiment to train a larger 152 layer Resnet for Imagenet. These results were obtained by splitting the computation across two GPUs. The relative accuracy results were similar to the 34-layer version, with 10-crop Top-5 error rates being [Baseline: 7.2%], [8-bit: 7.7%], and [4-bit: 7.7%].
>
> While our approximation method was able to fit the entire batch of 256 on two GPUs (128 on each), for the baseline we again had to do two forward-backward passes and average gradients (with 64 on each GPU in each pass). In this case too, we saw an advantage in wall-clock time because a batch of 64 for the baseline wasn't able to saturate all cores on each GPU. Our method took 17 seconds per iteration for the full batch (1 pass parallelized over two GPUs), while baseline training took 20 seconds (total of 2 passes over two GPUs).

---

### Official Review · AnonReviewer3 · 2018-11-09
**Clear description , lacking in comparison with relevant prior work.**

**Rating:** 5
**Confidence:** 3

**Review:**

In this paper the authors describe a quantization approach for activations of the neural network computation to improve the memory efficiency of neural network training and thus training efficiency of a single worker.

Prior work
-----------------
They compare the proposed method with other approaches involving the quantization of gradients or recomputation of activations in a sub-graph during back-propagation. However the literature survey lacks survey of more relevant quantization techniques e.g. [1].
[1] : Hubara, Itay, et al. "Quantized neural networks: Training neural networks with low precision weights and activations." The Journal of Machine Learning Research 18.1 (2017): 6869-6898.

experimental setup
-----------------------------
A more formal description of experimental setup assuming a general reader not familiar with the specific toolkits is advised. Any toolkit specific details like how the layer-wise forward & backward propagation is done via separate sess.run calls can be delegated to an appendix or footnote. Further given that the authors have chosen not to utilize the auto-diff functionality or other computation graph optimization features provided by Tensorflow; and given that they are even manually managing the memory allocation it is not clear why they are relying on this toolkit.  Irrespective of this choice, this section could be re-written to make the implementation description more accessible to a general reader and toolkit specific details could specified separately.

Reg. manual memory management - The authors specify how common buffers are being used for storing activations and gradients across layers. Given that typical neural network models need not be composed of homogenous layer types which can actually share the buffers it would be useful to add a detail on how much efficiency is achieved by reducing the memory allocation calls for the architectures being used in this paper.


results
-----------
Comparisons with prior work using other quantization methods to achieve memory efficiency is lacking.

---

> ### Author Response · Authors · 2018-11-11
> **Response**
>
> Thank you for the review. Please find answers to specific concerns below:
>
> - Regarding the Hubara et al. paper and comparisons to it:
>
> Hubara et al. address a very different problem than we do. Crucially, their method does not reduce memory usage **during training**, which is the goal of our work. Instead, they reduce the amount of memory and computation the network would need for inference (i.e., after training).
>
> Hubara et al.’s goal is to enable the use of quantized weights and activations at “test time” to reduce memory usage and computation cost in deployed networks. They train networks that can work with binary activations during inference, because it reduces model size and saves computation by turning floating point multiplications into binary operations. The paper addresses the challenge of how to train such quantized models, even though they are technically non-differentiable.
>
> Their approach provides no memory advantage during training itself (unlike us, this is not their goal). This is because their training method still relies on full-precision real-valued versions of the weights and activations, with discretization interspersed to match test time performance. Specifically, “Algorithm 1” in their paper clearly describes how their backward computation uses the real-valued versions of their binarized weights and activations. These are stored in memory at full precision during training.
>
> Our paper has a different goal: to train standard network models that will be used with full precision activations and weights for inference, using approximations to reduce their memory footprint during training. This is useful as training requires substantially more memory than inference, especially for deeper networks, due to the need for storing all intermediate activations. To clarify this, we will add a discussion of the Hubara et al. paper in our related work section.
>
> - Regarding Description of Experimental Setup: We will adopt the reviewer's suggestion and split the description of the implementation. We will first describe the general approach, and later specify the relationship to the Tensorflow toolkit. We note that we rely on Tensorflow simply as a matter of convenience. We use it because it allows us to call the efficient GPU routines for per-layer forward and gradient computation.
>
> - Regarding memory management: There is typically no loss or gain in efficiency due to memory allocation calls since these allocation calls are made once at the beginning of training and not at every iteration. This is true for our implementation, as well as regular training in most toolkits (including Tensorflow). This is because the structure of the network does not change from iteration to iteration, and so the toolkit is able to allocate all required buffers a-priori (or during the first iteration). Thus the main advantage of our method is in the reduction of total allocated memory.

---

### Author Response · Authors · 2018-11-27
**Revision**

We have uploaded a revised version of the paper incorporating the comments received so far by the revision deadline. We are of course happy to continue to respond to any further comments and questions.

We have responded to individual reviewers below. Here is a brief summary:

-Rev 1 has a positive view of our paper and suggested it could be further improved with experiments that illustrate the tangible benefits of our approach. Accordingly, we have added experiments to show the much larger batch-size practically allowed by our method, and the corresponding benefits to computational efficiency from better utilizing available parallel cores on a GPU.

-Rev 2’s main concern is that given the memory cost savings one gets from checkpointing, our method may not be practically needed. We have noted that our method can be used not just independently (it has lower computational overhead than the cost of a forward pass for checkpointing) but also along with checkpointing---since these are independent and complementary strategies for reducing memory use. (We have clarified this in the revision).

While check-pointing has a sub-linear (sqrt) memory cost wrt the number of layers, incorporating our method provides a factor improvement on that asymptotic cost (sqrt(An) vs sqrt(n) for A =⅛ or ¼).  We believe that there are many cases when checkpointing alone is not sufficient and further memory savings are needed, especially when a network is large not just because of depth (n), but also because of per-layer size: e.g., fully-convolutional networks with high-resolution images.

-Rev 3 asked about our relationship to Hubara et al’s work which also deals with quantization. We have clarified that their work does not reduce memory usage during training, but instead, seeks to reduce memory required for inference. We have included this in the revised related work section.

We have also adopted Rev 3’s suggestion of moving tensorflow-specific implementation details to an appendix.

---

### Meta-Review · Area_Chair1 · 2018-12-13

**Confidence:** 2
**Recommendation:** Reject

**Metareview:**

This work proposes to reduce memory use in network training by quantizing the activations during backprop. It shows that this leads to only small drops in accuracy for resnets on CIFAR-10 and Imagenet for factors up to 8. The reviewers raised concerns about comparison to other approaches such as checkpointing, and questioned the technical novelty of the approach.  The authors were able to properly address the concerns around comparisons, but the issue around novelty remained. This could be compensated by strengthening the experimental results and leveraging the memory saving for instance to train larger networks. Resubmission is encouraged.